# Lifespan estimation in marine turtles using genomic promoter CpG density

**Benjamin Mayne**[1]*, **Anton D. Tucker**[2], **Oliver Berry**[1], **Simon Jarman**[3]

**1** Environomics Future Science Platform, Indian Oceans Marine Research Centre, Commonwealth Scientific and Industrial Research Organisation, Crawley, Western Australia, Australia, **2** Department of Biodiversity, Conservation and Attractions, Marine Science Program, Kensington, Western Australia, Australia, **3** School of Biological Sciences, University of Western Australia, Perth, Western Australia, Australia

* benjamin.mayne@csiro.au

**Data Availability Statement:** All relevant data are within the paper and its Supporting Information files.

**Funding:** This project is supported by the North West Shelf Flatback Turtle Conservation Program

## Abstract

Maximum lifespan for most animal species is difficult to define. This is challenging for wildlife management as it is critical for estimating important aspects of population biology such as mortality rate, population viability, and period of reproductive potential. Recently, it has been shown cytosine-phosphate-guanine (CpG) density is predictive of maximum lifespan in vertebrates. This has made it possible to predict lifespan in long-lived species, which are generally the most intractable. In this study, we use gene promoter CpG density to predict the lifespan of five marine turtle species. Marine turtles are a particularly difficult group for lifespan estimation because of their migratory behaviour, longevity and high juvenile mortality rates, which all restrict individual tracking over their lifespan. Sanger sequencing was used to determine the CpG density in selected promoters. We predicted the lifespans for marine turtle species ranged from 50.4 years (flatback turtle, *Natator depressus*) to 90.4 years (leatherback turtle, *Dermochelys coriacea*). These lifespan predictions have broad applications in marine turtle research such as better understanding life cycles and determining population viability.

## Introduction

Marine turtles are slow growing, long-lived, and migrate vast distances in the ocean [1]. This makes it difficult to determine demographic characteristics of wild populations. Although mark-recapture studies of marine turtles can determine certain features of populations such as survival probabilities, it is difficult to determine the full extent of life cycles [2]. Consequently, making broader predictions relating to the risk of extinction, population growth, and viability due to limited age and longevity data is challenging [3]. Although marine turtles are known to be long-lived, the true longevity of each species is unknown [4]. Marine turtles epitomise the difficulties in generating lifespan or longevity information in many wild animal species.

Lifespan is difficult to define for most species of animals, especially in long-lived species, which may outlive a generation of researchers. Lifespan is commonly regarded as being the highest recorded age of an individual or the age at death within a selected population [5]. Lifespan is an essential characteristic for any species and has implications for wildlife

and the CSIRO Environomics Future Science Platform. The funders had no role in study design, data collection and analysis, decision to publish, or preparation of the manuscript.

**Competing interests:** The authors have declared that no competing interests exist.

management. Lifespan is associated with life-history traits such as reproductive capacity and the probability of mortality [6]. Currently, of the seven marine turtle species that occur globally, only the Green sea turtle (*Chelonia mydas*) has a reliable lifespan value in the Animal Ageing & Longevity Database (An Age; 75 years) [7–9]. Lifespan predictions for the remaining species is typically based on a small number of *ad hoc* and opportunistic records, often for animals held in captivity [10]. This limits the number of analyses relating to population growth and viability that can be performed for the other marine turtles as they require longevity data [3].

Previous research has found the frequency of cytosine-phosphate-guanine (CpG) sites in selected gene promoters can be predictive of lifespan [11, 12]. This provides an alternative method to predicting lifespan in long-lived species. Here, we predict the lifespan of marine turtle species that occur in Australian waters using CpG density in gene promoters. The molecular lifespan predictions provided in this study have broad application in the wildlife management of marine turtles.

## Materials and methods

### Animal ethics

Animal ethics for the collection of tissue was approved by the Department of Biodiversity, Conservation and Attractions (FO25000245).

### Tissue collection and DNA extraction

Tissue was collected from one individual of each species (Table 1). Flipper biopsies from marine turtles were stored in 70% ethanol. DNA was extracted from tissue using the DNeasy Blood & Tissue Kit (QIAGEN) following the manufacture's protocol. DNA was quantified using a QIAxpert (QIAGEN).

### PCR design and sanger sequencing

Since the five marine turtles of interest do not have published genomes, we used the green sea turtle genome (CheMyd 1.0) as a reference genome [13]. The green sea turtle is the only marine turtle with a reference genome available. The lifespan promoters were identified using Basic Local Alignment Search Tool (BLAST) v2.2.31 (S1 Appendix) [14]. Primers were designed using Primer3 v0.4.0 for an optimal primer length of 20bp and temperature of 60˚C [15]. A temperature gradient (45–60˚C) was used for each primer pair to determine the optimal annealing temperature in each species (S2 Appendix). PCR reactions that produced single band visualised on an agarose gel were used for Sanger sequencing (Australian Genome Research Facility). Promoter CpG density was determined by calculating the CpG frequency within the BLAST hit based on the Green Sea Turtle genome and dividing it by the BLAST hit length (bp).

**Table 1. Locations of sea turtles where tissue was collected for DNA extraction.**

| Species | Location | Latitude | Longitude |
|---|---|---|---|
| Leatherback sea turtle (*Dermochelys coriacea*) | Albany, Western Australia | -30.505 | 115.066 |
| Loggerhead sea turtle (*Caretta caretta*) | South Muiron Islands, Western Australia | -25.498 | 112.987 |
| Olive Ridley sea turtle (*Lepidochelys olivacea*) | Roebuck Bay, Western Australia | -18.019 | 122.237 |
| Hawksbill sea turtle (*Eretmochelys imbricata*) | Delambre Island, Western Australia | -20.451 | 117.076 |
| Flatback sea turtle (*Natator depressus*) | Roebuck Bay, Western Australia | -18.036 | 122.284 |
| Green sea turtle (*Chelonia mydas*) | South Muiron Islands, Western Australia | -20.425 | 115.591 |

### Predicting lifespan

Lifespan prediction was determined using the model developed previously that is exclusive to five vertebrate classes [12]. Lifespan prediction is conducted by determining the CpG density within selected genomic promoters. Genomic promoters predictive of lifespan were identified by comparing the sequences of 29,598 promoters to a database of known animal lifespans [7, 16]. An elastic net regression model was used to regress the lifespans of 252 species against the CpG densities of the genomic promoters. The model returned a total of 42 genomic promoters and coefficients that can be used to predict lifespan. The model returns the most informative genomic promoters but does allow redundancy as not all species will contain all 42 genomic promoters. The model was found to have an error range of 5.9%. The model returns a single lifespan prediction, but the 5.9% error is given in ± years. CpG densities were calculated for each promoter that received a BLAST hit to the Green Sea Turtle genome. A significant BLAST hit in the green sea turtle genome was considered with an identity > 70%.

### Average mass and length data

To determine whether basic morphological traits correlated with predicted lifespans, physical features including the average carapace length (mm) and mass (g) for each species was obtained from the Animal Diversity Web (ADW) database [17]. Olive ridley sea turtles did not have data available in the ADW database and were removed from the analysis. Pearson correlations between physical features and maximum lifespan were natural log (ln) transformed to determine if there was a linear relationship. All analyses were performed in R v3.5.1 [18].

## Results

### Lifespan prediction

Promoter CpG density used for lifespan prediction is provided in S3 Appendix. The lifespan prediction for five marine turtle species are detailed in Table 2. The green sea turtle was excluded from the analysis as it has a known lifespan [7–9]. Leatherback sea turtles were found to have the longest lifespan prediction at 90.4 ± 5.3 years and flatback sea turtles with the shortest at 50.4 ± 2.9 years.

### Lifespan and physical features

We found a strong positive correlation between both the average length (cor = 0.95, p-value = 0.012) and mass (cor = 0.98, p-value = 0.0038) with the lifespan prediction from CpG densities in marine turtles (Fig 1). Positive correlations were also observed in untransformed data for both length (cor = 0.91, p-value = 0.030) and mass (cor = 0.96, p-value = 0.010).

## Discussion

Marine turtles globally face many anthropogenic threats [19]. However, as with other long-lived organisms their lifespan is difficult to determine and data on this key life-history attribute

**Table 2. Lifespan prediction of marine turtle species using promoter CpG density.**

| Species | Prediction (- 5.9% Error) | Prediction | Prediction (+ 5.9% Error) |
|---|---|---|---|
| Leatherback sea turtle (*Dermochelys coriacea*) | 85.1 | 90.4 | 95.7 |
| Loggerhead sea turtle (*Caretta caretta*) | 59.1 | 62.8 | 66.5 |
| Olive Ridley sea turtle (*Lepidochelys olivacea*) | 51.1 | 54.3 | 57.5 |
| Hawksbill sea turtle (*Eretmochelys imbricata*) | 50.1 | 53.2 | 56.4 |
| Flatback sea turtle (*Natator depressus*) | 47.4 | 50.4 | 53.4 |

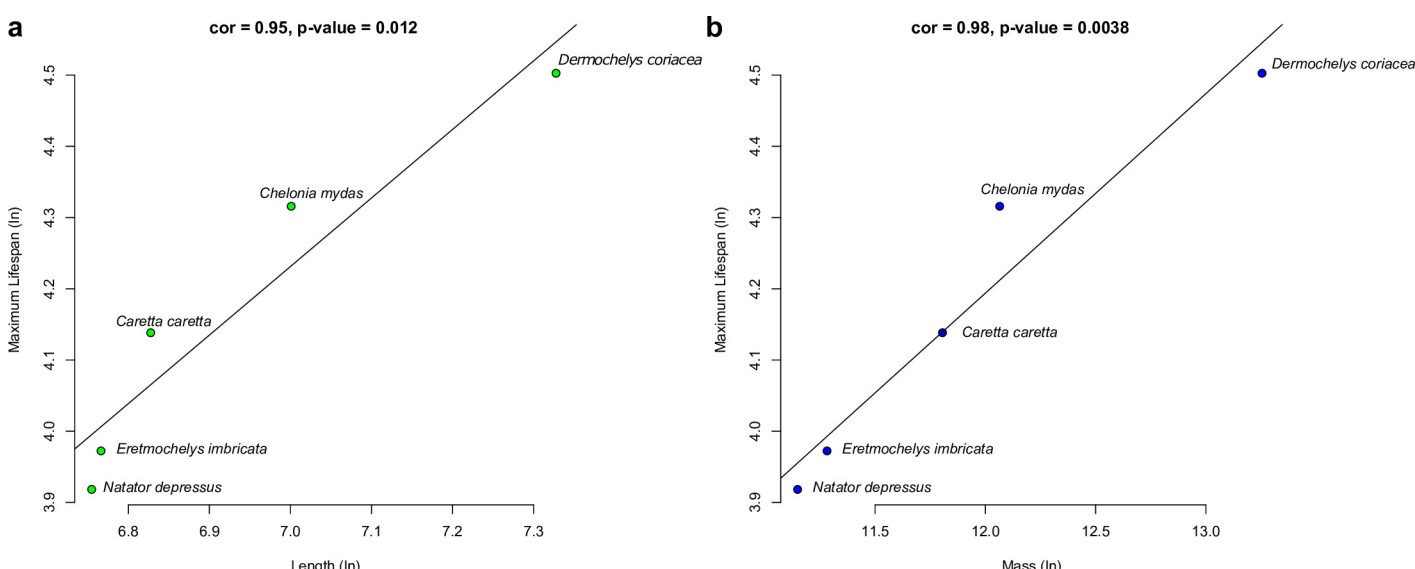

**Fig 1.** Increasing **a.** carapace length and **b.** mass of marine turtle species with lifespan. Each dot represents a species. Average length and mass data was obtained from the Animal Diversity Web database [17].

is sparse. This may partly be attributed to the fact that they may out-live research projects or researchers themselves. Age-estimates for marine turtles do exist but are based on much weaker data than typically is available for short-lived species. A lifespan prediction, provides an immediate value thereby providing useful demographic parameter regarding marine turtle ecology. In this study, we have used a molecular approach to confirm marine turtles as being long-lived animals. We found the leatherback sea turtle to have the longest lifespan and the flatback sea turtle with the shortest, with a difference of 40 years. This suggests a high variance and specific lifespan between species. The lifespan predictions provide a fundamental parameter used in determining mortality rates [20]. This can be used in the wildlife management of marine turtles and determine if specific populations are at risk of extinction.

Reliable lifespan values for long-lived species are difficult to find within the literature, although some do exist for selected individuals. Leatherback sea turtles were found to have the longest lifespan at 90 years. They have been reported to live at least 30 years in the wild with informal evidence suggesting a longevity of 70–80 years [21]. Loggerhead sea turtles have also been reported to have a lifespan of at least 30 years and up to 60 years in the wild [22, 23]. Similarly, the Olive Ridley, Hawksbill, and Flatback sea turtles have had reported lifespans of at least 30 years and up to 50 years in the wild [24, 25]. These reported lifespan values are supportive of the molecular predictions. A limitation of these studies is the low samples size as they only followed selected individuals. The longevity of marine turtle's life cycles makes it challenging to study and determine the maximum lifespan.

Age estimates of wild animals can provide insight into age at sexual maturation and longevity [26, 27]. Skeletochronology is used to determine the age of stranded deceased marine turtles [28, 29]. Previous studies have found, depending on the species, that the age at sexual maturity ranges from as early as 6 years (Kemp's ridley sea turtle) to 35 years (leatherback sea turtle) [30–33]. Other studies researching the same species, but different populations have recorded different ages at sexual maturity. For example with loggerhead sea turtles age at sexual maturation can range from 20 years of age in North American populations to 35 years in Australia [34–36]. Although longevity can be determined from age at sexual maturation, it can range greatly between different populations. Older age at sexual maturation ranges suggest marine

turtles are long-lived animals. Other studies have found wild sea turtles of at least 40 years of age [28, 37]. Many of these age estimates are on the lower end of the lifespan predictions presented in this paper. Older individuals may exist in the wild and are not recorded since skeletochronology can only be carried out on deceased individuals. Therefore, the lifespan predictions provide a useful but potentially conservative values. However, it is important to note that older sea turtles are known to exist, primarily in captivity. It is well known animals, including reptiles, which are kept in captivity generally live longer than their wild counterparts [38, 39]. The lifespan predictions presented here suggest they may be on the upper end of what can be achieved in the wild but may be considered low to what can occur in captivity.

We found two morphological metrics of turtle size to strongly correlate with increasing lifespan (Fig 1). When more data becomes available the loggerhead and Kemp's ridley sea turtle morphological and lifespan data can be added to the analyses. As a life-history strategy this may reflect a lower death rate in larger animals from extrinsic causes such as predation [40]. This may be the case with marine turtles since except for humans and large sharks, adults have few predators [41]. This correlation between size and longevity is well established in other taxa, supporting our findings in marine turtles [42]. To our knowledge this is the first time that this has been demonstrated in marine turtles, an ancient vertebrate group [42].

The main limitation of using a molecular method to predict lifespan is the generalisation of the species. A single molecular prediction does not account for population differences. Environmental pressures differ between populations which may reduce life expectancies. Without factoring environmental pressures, the molecular method cannot be used to make predictions for specific populations or individuals. Rather, it represents a potential maximum lifespan for the species. A maximum lifespan can be used as a reference tool see if individuals within a population are reaching their natural limit. If their life expectancy is low compared to their maximum lifespan it may indicate a potential environmental factor that may be limiting their longevity. Another limitation is the lack of known age data. Skeletochronology is used to determine the age of turtles but by having a non-invasive method, older aged turtles can be determined. This can then be used to determine if some turtles are either approaching or exceeding the lifespan predictions in this paper. A limitation of the method used in this study to predict lifespan is the dependency on an assembled genome. Reference genomes are in different stages of assembly such as contigs, scaffolds, or at the chromosome level. This can introduce artefacts and may result in inaccurate CpG densities. In this study, sanger sequencing was used to determine CpG density thereby removing the possibility of a lack of coverage. Lifespan prediction from DNA has shown to be highly predictive across most speciose vertebrate classes, including reptilia [12]. In the absence of robust observational information on the lifespans of wild marine turtles, molecular predictions represent useful consistently derived foundation values for this iconic and vulnerable group of marine animals.

## Supporting information

**S1 Appendix. Green sea turtle genomic coordinates and primer sequences used to amplify promoter sequences.**
(XLSX)

**S2 Appendix. Species specific annealing temperatures for each primer pair.**
(XLSX)

**S3 Appendix. Promoter CpG density from sanger sequencing used to predict marine turtle lifespan.**
(XLSX)

## Acknowledgments

The authors would like to thank all Department of Biodiversity, Conservation and Attractions (DBCA) researchers who were involved in the collection of marine turtle tissue which was used in this study. The authors would also like to thank the two reviewers' suggestions to improving the manuscript.

## Author Contributions

**Conceptualization:** Benjamin Mayne, Simon Jarman.

**Data curation:** Benjamin Mayne.

**Formal analysis:** Benjamin Mayne.

**Funding acquisition:** Oliver Berry.

**Investigation:** Anton D. Tucker.

**Methodology:** Benjamin Mayne, Simon Jarman.

**Resources:** Anton D. Tucker.

**Writing – original draft:** Benjamin Mayne, Simon Jarman.

**Writing – review & editing:** Benjamin Mayne, Anton D. Tucker, Oliver Berry, Simon Jarman.

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
