## [Decision Letter · Decision Letter 0]

30 Jun 2020

PONE-D-20-15932

Lifespan estimation in marine turtles using genomic promoter CpG density

PLOS ONE

Dear Dr. Mayne,

Thank you for submitting your manuscript to PLOS ONE. After careful consideration, we feel that it has merit but does not fully meet PLOS ONE’s publication criteria as it currently stands. Therefore, we invite you to submit a revised version of the manuscript that addresses the points raised during the review process.

There are several clarifications needed to ensure your study is understood to present a method that allows the prediction of life span, and not estimates based on previously obtained date. There is a significant difference between these two. Additionally, some of the descriptions require more detail to make them useful. Please refer to individual reviewer's comments.

We look forward to receiving your revised manuscript.

Kind regards,

Ulrike Gertrud Munderloh, Ph.D.

Academic Editor

PLOS ONE

Journal Requirements:

Reviewers' comments:

Reviewer's Responses to Questions

**Comments to the Author**

1. Is the manuscript technically sound, and do the data support the conclusions?

Reviewer #1: Partly

Reviewer #2: Yes

2. Has the statistical analysis been performed appropriately and rigorously? 

Reviewer #1: I Don't Know

Reviewer #2: Yes

3. Have the authors made all data underlying the findings in their manuscript fully available?

Reviewer #1: Yes

Reviewer #2: Yes

4. Is the manuscript presented in an intelligible fashion and written in standard English?

Reviewer #1: Yes

Reviewer #2: Yes

5. Review Comments to the Author

Reviewer #1: For long-lived species, it is difficult or impossible for human observers to accurately estimate maximum lifespan through direct observation, leaving ecologists & other biologists with an important missing datum for their species of interest. In this manuscript, Mayne et al take advantage of a published phenomenon for vertebrates – the correlation between CpG density in a select set of gene promoters and observed maximal lifespan – to predict the lifespans for several species of long-lived marine turtles. This manuscript develops no method nor tests a hypothesis; instead, it provides the only source of maximum-lifespan data currently available for these species. And due to the impracticality of direct observation, it is unlikely that an alternative, direct observation will ever be produced. This manuscript will likely therefore be a valuable reference for those who study marine turtles.

The main problem with this manuscript would be easily remedied with text edits: the authors generally describe the results of their analysis as “lifespans” or “estimated lifespans”. This diction inaccurately reflects the nature of the manuscript: “estimate” suggests an imprecise but nonetheless observation-based measurement. There are NO observed lifespans in this manuscript, just predictions. As I’ve said above, that’s fine, and there is great value in these predictions. But it is important that the authors make that abundantly clear throughout, so that it will be obvious to even the most casual reader. In their prior publication, where they established this correlative phenomenon to hold across vertebrates (listed as ref. 12), these authors were very good about using the diction of “predictied lifespan”. They should maintain that diction in this manuscript.

A more specific example of my comment above: the y-axis labels in Fig 1 (and descriptive text on lines 90-3). The figure represents the correlation between actual data collected in this paper versus literature values for length & weight. It would be more appropriate to label the Y-axis in terms of the observed CpG density values (the actual data!) than in terms of a value inferred based on those densities.

There are a few places where this manuscript is overly vague, and some additional clarification should be added. In Table 2, the statistical nature of the “lower bound” and “upper bound” intervals is unclear and should be specified. Ditto for the “+/-“ qualifiers on line 88 of the Results. Are these all based on the 5.9% error for the correlation in reference 12? Or are statistics for these particular measurements of CpG density also taken into account? Secondly, the methods for “Lifespan estimation” should be somewhat expanded. I appreciate that published methods should only require brief summaries, but this is too brief given how central this computational method is to the manuscript. The authors cite reference 12, which in turn cites reference 11. The summary following “Briefly, as described previously…” in reference 12 should serve as a guide to the authors as to the level of detail that is needed here.

Reviewer #2: The authors present lifespan estimates for five sea turtle species which are noted for their longevity and with life history traits making experimental determination of lifespan intractable. The method used is based solely on CpG density and is applied here in non-model organisms for which only one species has a fully sequenced genome. Therefore, this work demonstrates a number of novel and useful discoveries, 1) lifespan can be estimated for wild populations that are unable to be tracked individually, 2) a subset of promoter sequences can be sequenced to estimate lifespan without full genome sequencing, 3) lifespan can be calculated in marine vertebrates which are not closely related to other species with known lifespans, and 4) only small tissue samples from individuals at any age are necessary to calculate lifespan, negating the need to keep animals in captivity or sacrificial tissue collection. While the authors had previous experience in generating lifespan data for species with whole genomes available, this is the first report of species lifespan estimates being generated using de novo Sanger sequencing from tissue in non-sequenced animals. The authors also report the first link between size and longevity in marine turtles, a feature seen in many terrestrial species but never before confirmed in marine turtles. These lifespan estimates will be particularly useful, the authors note, in ecological studies of population age structure to determine whether ecological pressures are limiting populations from reaching their natural lifespans.

I note that the lifespan estimates do not uniformly correspond to phylogeny. For example, Lepidochelys olivacea and Caretta caretta are most closely related, and have similar estimates (62 and 54 years), while Natator depressus is most closely related to the green sea turtle Chelonia mydas, yet their lifespans are quite divergent, 50 and 75 years respectively. These differences suggest that these species have probably been under diverging selection pressures, at least regarding lifespans, although a 20-30 million range since speciation is a long time to collect differences.

Overall the authors did an outstanding job and provided novel, useful age estimates for a vulnerable keystone species with previously unreliable age estimates. This paper is concise, well-presented, and serves as a guide for others to report future molecular based lifespan estimates.

Minor concerns:

Though I recommend “Accept”, the authors should still briefly summarize the model used for the prediction and how it was used with the density data determined here in the methods section lines 72-73. It’s a bit too brief in this manuscript.

Calculating CpG density by dividing CpG frequency by the BLAST hit length is acceptable when using a single reference genome with uniform coverage and relatively closely related species. However, there is a risk of artefacts if this method is expanded using multiple reference genomes that may have disparate quality or coverage, leading to variable length BLAST hits for similar promoter regions. This concern is not applicable here due to study design but may be an issue that could be addressed by determining a consistent length or algorithm for CpG density measures in genomic regions in future studies.

Reference 6 has a URL attached that does not seem correct.

In the results, line 87, the green sea turtle lifespan refers to ref #6 and should probably be ref #7-9 or another.

6. PLOS authors have the option to publish the peer review history of their article (what does this mean?). If published, this will include your full peer review and any attached files.

Reviewer #1: No

Reviewer #2: **Yes: **Christopher Faulk

---

## [Author Response · Author response to Decision Letter 0]

13 Jul 2020

Review Comments to the Author

Reviewer #1: For long-lived species, it is difficult or impossible for human observers to accurately estimate maximum lifespan through direct observation, leaving ecologists & other biologists with an important missing datum for their species of interest. In this manuscript, Mayne et al take advantage of a published phenomenon for vertebrates – the correlation between CpG density in a select set of gene promoters and observed maximal lifespan – to predict the lifespans for several species of long-lived marine turtles. This manuscript develops no method nor tests a hypothesis; instead, it provides the only source of maximum-lifespan data currently available for these species. And due to the impracticality of direct observation, it is unlikely that an alternative, direct observation will ever be produced. This manuscript will likely therefore be a valuable reference for those who study marine turtles.

The main problem with this manuscript would be easily remedied with text edits: the authors generally describe the results of their analysis as “lifespans” or “estimated lifespans”. This diction inaccurately reflects the nature of the manuscript: “estimate” suggests an imprecise but nonetheless observation-based measurement. There are NO observed lifespans in this manuscript, just predictions. As I’ve said above, that’s fine, and there is great value in these predictions. But it is important that the authors make that abundantly clear throughout, so that it will be obvious to even the most casual reader. In their prior publication, where they established this correlative phenomenon to hold across vertebrates (listed as ref. 12), these authors were very good about using the diction of “predictied lifespan”. They should maintain that diction in this manuscript.

Response: As the reviewer points out there are no observed lifespans for marine turtles and therefore the use of estimated lifespan in the manuscript is inaccurate and can cause confusion. To amend the problem the term estimated lifespan has been replaced with predicted lifespan.

A more specific example of my comment above: the y-axis labels in Fig 1 (and descriptive text on lines 90-3). The figure represents the correlation between actual data collected in this paper versus literature values for length & weight. It would be more appropriate to label the Y-axis in terms of the observed CpG density values (the actual data!) than in terms of a value inferred based on those densities.

Response: The reviewer makes an important point in regards to the Y-axis labels in Figure 1. We have now modified the Y-axis labels from “Maximum Lifespan” to “Lifespan Prediction from CpG density”. The text on lines 90-3 have also been updated to reflect the Y-axis label changes.

There are a few places where this manuscript is overly vague, and some additional clarification should be added. In Table 2, the statistical nature of the “lower bound” and “upper bound” intervals is unclear and should be specified. Ditto for the “+/-“ qualifiers on line 88 of the Results. Are these all based on the 5.9% error for the correlation in reference 12? Or are statistics for these particular measurements of CpG density also taken into account? Secondly, the methods for “Lifespan estimation” should be somewhat expanded. I appreciate that published methods should only require brief summaries, but this is too brief given how central this computational method is to the manuscript. The authors cite reference 12, which in turn cites reference 11. The summary following “Briefly, as described previously…” in reference 12 should serve as a guide to the authors as to the level of detail that is needed here.

Response: The reviewer makes an important point regarding being too brief in the methods and the model. 

In table 2 we have changed the headings to Prediction ± 5.9% and on lines 72-81 we have provided more detail on the model that was used for lifespan prediction. We now explain that the ± qualifiers are representing the 5.9% error in years. We also provide a brief discussion in the methods on how the model was initially concepted, build and how it can be applied on other species. 

Reviewer #2: The authors present lifespan estimates for five sea turtle species which are noted for their longevity and with life history traits making experimental determination of lifespan intractable. The method used is based solely on CpG density and is applied here in non-model organisms for which only one species has a fully sequenced genome. Therefore, this work demonstrates a number of novel and useful discoveries, 1) lifespan can be estimated for wild populations that are unable to be tracked individually, 2) a subset of promoter sequences can be sequenced to estimate lifespan without full genome sequencing, 3) lifespan can be calculated in marine vertebrates which are not closely related to other species with known lifespans, and 4) only small tissue samples from individuals at any age are necessary to calculate lifespan, negating the need to keep animals in captivity or sacrificial tissue collection. While the authors had previous experience in generating lifespan data for species with whole genomes available, this is the first report of species lifespan estimates being generated using de novo Sanger sequencing from tissue in non-sequenced animals. The authors also report the first link between size and longevity in marine turtles, a feature seen in many terrestrial species but never before confirmed in marine turtles. These lifespan estimates will be particularly useful, the authors note, in ecological studies of population age structure to determine whether ecological pressures are limiting populations from reaching their natural lifespans.

I note that the lifespan estimates do not uniformly correspond to phylogeny. For example, Lepidochelys olivacea and Caretta caretta are most closely related, and have similar estimates (62 and 54 years), while Natator depressus is most closely related to the green sea turtle Chelonia mydas, yet their lifespans are quite divergent, 50 and 75 years respectively. These differences suggest that these species have probably been under diverging selection pressures, at least regarding lifespans, although a 20-30 million range since speciation is a long time to collect differences.

Overall the authors did an outstanding job and provided novel, useful age estimates for a vulnerable keystone species with previously unreliable age estimates. This paper is concise, well-presented, and serves as a guide for others to report future molecular based lifespan estimates.

Minor concerns:

Though I recommend “Accept”, the authors should still briefly summarize the model used for the prediction and how it was used with the density data determined here in the methods section lines 72-73. It’s a bit too brief in this manuscript.

Response: Reviewer 2 shares the same concern as reviewer 1 regarding the methods section as being to brief. The model is described in more detail on lines 72-81.

Calculating CpG density by dividing CpG frequency by the BLAST hit length is acceptable when using a single reference genome with uniform coverage and relatively closely related species. However, there is a risk of artefacts if this method is expanded using multiple reference genomes that may have disparate quality or coverage, leading to variable length BLAST hits for similar promoter regions. This concern is not applicable here due to study design but may be an issue that could be addressed by determining a consistent length or algorithm for CpG density measures in genomic regions in future studies.

Response: The reviewer makes an important point regarding reference genomes with disparate quality or coverage. This would be difficult to incorporate into a model as genome coverage would be factor of both genome size and sequencing depth. It would be ideal to only use the model to predict lifespan using fully assembled genomes. We have now provided this discussion on lines 168 -172, to urge the reader that the method is ideal to fully assembled genomes. 

Reference 6 has a URL attached that does not seem correct.

In the results, line 87, the green sea turtle lifespan refers to ref #6 and should probably be ref #7-9 or another.

Response: Ref 6 was incorrect and has been replaced with correct refs 7-9 as the reviewer has pointed out.

---

## [Editor Report · Decision Letter 1]

16 Jul 2020

Lifespan estimation in marine turtles using genomic promoter CpG density

PONE-D-20-15932R1

Dear Dr. Mayne,

We’re pleased to inform you that your manuscript has been judged scientifically suitable for publication and will be formally accepted for publication once it meets all outstanding technical requirements.

Kind regards,

Ulrike Gertrud Munderloh, Ph.D.

Academic Editor

PLOS ONE
---

## [Editor Report · Acceptance letter]

21 Jul 2020

PONE-D-20-15932R1 

Lifespan estimation in marine turtles using genomic promoter CpG density 

Dear Dr. Mayne:

I'm pleased to inform you that your manuscript has been deemed suitable for publication in PLOS ONE. Congratulations! Your manuscript is now with our production department. 

Kind regards, 

on behalf of

Dr. Ulrike Gertrud Munderloh 

Academic Editor

PLOS ONE